# Morphological and molecular identification of the dioecious "African species *Volvox rousseletii* (Chlorophyceae) in the water column of a Japanese lake based on field-collected and cultured materials

**Ryosuke Kimbara**[1], **Nanako Isaka**[1], **Ryo Matsuzaki**[2,3], **Hiroko Kawai-Toyooka**[1], **Masanobu Kawachi**[2], **Hisayoshi Nozaki**[1] *

**1** Department of Biological Sciences, Graduate School of Science, The University of Tokyo, Hongo, Bunkyo-ku, Tokyo, Japan, **2** Center for Environmental Biology and Ecosystem Studies, National Institute for Environmental Studies, Onogawa, Tsukuba, Ibaraki, Japan, **3** Faculty of Life and Environmental Sciences, University of Tsukuba, Tennodai, Tsukuba, Ibaraki, Japan

* nozaki@bs.s.u-tokyo.ac.jp

**Data Availability Statement:** New sequence data, alignments used for our phylogenetic analyses, and

## Abstract

*Volvox rousseletii* is a dioecious species belonging to *Volvox* sect. *Volvox* that has previously only been found in Africa. During field surveys in a large dam lake (Lake Sagami) in Kanagawa Prefecture, central Japan, we encountered a *Volvox* sect. *Volvox* species that produces dioecious sexual spheroids in the water column. Although sexual induction of this species in culture did not produce adequately well-developed sexual spheroids for species identification, molecular data directly obtained from field-collected sexual spheroids verified the identity of field-collected male and female sexual spheroids as well as cultured materials. Based on molecular and morphological data, the species was identified as *V. rousseletii*. This is the first record of a dioecious species of *Volvox* sect. *Volvox* in Japan.

## Introduction

*Volvox* sect. *Volvox* is a morphologically interesting group of green algae due to the presence of thick cytoplasmic bridges between constitutive cells and spines on the zygote walls [1,2]. This section of *Volvox* includes seven monoecious and three dioecious species [3–5]. Although three of the monoecious species, *V. globator*, *V. barberi*, and *V. capensis*, are found on at least two different continents [1,3,4], records of the dioecious species of this section are not as frequent, and each has previously only been found in a small area of a single continent [1,5,6]. The three dioecious species are distinguished from one another primarily by differences in the morphology of the sexual spheroids [1,5]. In Japan, at least two monoecious species of *Volvox* sect. *Volvox* have been found [3,7]. However, dioecious species of this section have not previously been recorded from Japan.

new strains are available under the DDBJ/ENA/ GenBank accession numbers (LC493797– LC493811), TreeBASE ID (S24792), and NIES Collection strain designations (NIES-4336–NIES-4342), respectively. All other relevant data are within the paper and its Supporting Information files.

**Funding:** HN was supported by a Grant-in-Aid for Scientific Research (grant number 16H02518) from the Ministry of Education, Culture, Sports, Science and Technology (MEXT)/Japan Society for the Promotion of Science (JSPS) KAKENHI (https://www.jsps.go.jp/english/e-grants/). The funder had no role in study design, data collection and analysis, decision to publish, or preparation of the manuscript.

**Competing interests:** The authors have declared that no competing interests exist.

During our field surveys of large dam lakes in Kanagawa Prefecture, central Japan, we encountered dioecious sexual spheroids of *Volvox* sect. *Volvox* in the water column of a lake. Although a subsequent culture experiment did not produce adequately well-developed and fully mature sexual spheroids for species identification, molecular data obtained directly from the fully developed sexual spheroids collected from the lake water resolved the species identity. The species was identified as *V. rousseletii*, which has previously only been reported from Africa [1,6]. The morphology, phylogeny, and taxonomy of *V. rousseletii* are described in this report.

# Materials and methods

## Ethics statement

We collected *Volvox rousseletii* from the water column of a large dam lake, Lake Sagami. Collection locations and details are shown in Table 1. The collection of volvocalean algae in Lake Sagami and from water storage tanks of the Tanigahara Water Purification Plant was permitted by the Tanigahara Water Purification Plant of Kanagawa Prefecture Companies Authority, the Kanagawa Prefectural Government, Japan.

## Establishment of cultures and light microscopic observations

Water samples were collected from Lake Sagami and storage tanks of lake water from Lake Sagami within the Tanigahara Water Purification Plant (Table 1). Clonal cultures of *Volvox rousseletii* were established from the water samples in Petri dishes (90 × 20 mm), using the pipette-washing method [8]. The cultures were grown in screw-cap tubes (18 × 150 mm) containing 10–11 mL artificial freshwater-6 (AF-6) [9] or AF-6/3 medium (AF-6 medium diluted with two volumes of distilled water [4]) at 20˚C, 23˚C, or 25˚C on a 14 h light:10 h dark schedule under cool-white fluorescent lamps at an intensity of 80–130 $\mu$mol·m$^{-2}$·s$^{-1}$. Because the asexual spheroids of the dioecious species were indistinguishable from those of the monoecious species *Volvox* sp. Sagami in the same lake [7], clonal culture strains of the candidate dioecious species of *Volvox* sect. *Volvox* (*V. rousseletii*) were selected from the aforementioned established strains based on molecular information (see below). The new wild strains of the dioecious species *Volvox rousseletii* from Japan (Table 1) are available from the Microbial Culture Collection at the Institute for National Environmental Studies (NIES Collection, Tsukuba, Japan) [10] (http://mcc.nies.go.jp/index_en.html) as NIES-4336–NIES-4342 (Table 1).

To observe the morphology of asexual spheroids, the cultures were grown in *Volvox* thiamin acetate (VTAC) medium containing 200 mg L$^{-1}$ sodium acetate 4H$_2$O [10,11] or VTAC/3 (VTAC medium diluted with two volumes of distilled water [2]) at 25˚C on 14:10 LD. A small aliquot of asexual spheroids in actively grown 2- to 5-day-old cultures in tubes or Petri dishes (55 × 15 mm) was examined. Sexual spheroids did not develop spontaneously in culture with either VTAC or VTAC/3 medium. To induce production of sexual spheroids in culture, urea soil *Volvox* thiamin/3 (USVT/3) medium [7] (USVT medium [VTAC medium supplemented with 40 mg L$^{-1}$ urea and 40 mL L$^{-1}$ soil extract medium] [4] diluted with two volumes of distilled water) was also used, and these cultures were grown at 25˚C on 14:10 LD. To enhance sexual induction, 0.1–0.2 mL inducer (supernatant of the male culture after production of male sexual spheroids and sperm packets) was added to the USVT/3 medium and grown at 32˚C on 14:10 LD. For maturation of sexual spheroids, 0.5–1.0 mL actively growing culture with sexual spheroids was inoculated into 10–11 mL USVT/3 medium.

Light microscopy was performed using a BX60 microscope (Olympus, Tokyo, Japan) equipped with Nomarski optics. The number of cells in spheroids was counted as described previously [1,11]. Individual cellular sheaths of the gelatinous matrix of the spheroids were

**Table 1. List of species and field-collected samples/strains used in the present phylogenetic analyses (Figs 4 and 5).**

| Species | Sample/strain designation | Origin of sample/strain | GenBank/EMBL/DDBJ Accession number | | |
|---|---|---|---|---|---|
| | | | ITS-1, 5.8S rDNA and ITS-2 | *rbcL* | *psbC* |
| *Volvox rousseletii* from Japan | 2015-0610-3v6 [a,b] (= NIES-4336) | Water sample collected from Lake Sagami, Kanagawa, Japan (water temperature 21°C; pH 9.1; N 35° 36.625', E 139° 11.150') in 10 June 2015. | LC493797 [c] | LC493808 [c] | LC493810 [c] |
| | 2015-0610-3v7 [a,b] (= NIES-4337) | | LC493798 [c] | | |
| | 2015-0610-3v9 [a,b] (= NIES-4338) | | LC493799 [c] | | |
| | v-sgm-17 [a,b] (= NIES-4339) | Water sample collected from Lake Sagami, Kanagawa, Japan (water temperature 28°C; pH 9.6; N 35°36.732', E 139°11.457') in 26 July 2018. | LC493800 [c] | LC493809 [c] | LC493811 [c] |
| | v-sgm-23 [a,d] (= NIES-4340) | | LC493801 [c] | | |
| | v-sgm-24 [a,b] (= NIES-4341) | | LC493802 [c] | | |
| | Male-Sagami [e] | | LC493803 [c] | | |
| | FeEg-Sagami [f] | | LC493804 [c] | | |
| | v-tani-9 [a,b] (= NIES-4342) | Water sample collected from Tanigahara Water Purification Plant, Kanagawa, Japan (water temperature 22°C; pH 10.1; N 35° 35.459', E 139° 17.782') in 20 September 2018. | LC493805 [c] | | |
| | Male-Tani [e] | | LC493806 [c] | | |
| | FeZy-Tani [g] | | LC493807 [c] | | |
| *Volvox* sp. Sagami | NIES-4021 | Japan | LC191308 | LC191316 | LC191326 |
| *Volvox capensis* | M1-2 (= NIES-3874) | USA | LC0338704 | LC033870 | LC033872 |
| *Volvox kirkiorum* | NIES-2740 | Japan | AB663324 | AB663322 | AB663323 |
| *Volvox ferrisii* | NIES-2736 | Japan | AB663336 | AB663334 | AB663335 |
| *Volvox globator* | SAG 199.80 (= UTEX 955) | USA | AB663340 | D86836 | AB044478 |
| *Volvox barberi* | UTEX 804 | USA | AB663341 | D86835 | AB044477 |
| *Volvox rousseletii* from South Africa | UTEX 1862 (= NIES-734) | South Africa | AB663342 | D63448 | AB044479 |
| *Volvox perglobator* | Tucson | USA | MG429137 | | |
| | VspTf | | | KY489662 | KY489659 |
| *Colemano-sphaera angeleri* | NIES-3382 | Japan | | AB905592 | AB905598 |
| *Colemano-sphaera charkowiensis* | NIES-3383 | Japan | | AB905591 | AB905598 |
| *Platydorina caudata* | NIES-728 (= UTEX 1658) | USA | | D86828 | AB044494 |

[a] Established in this study.

[b] Male strain.

[c] Sequenced in this study.

[d] Female strain.

[e] Sexual male spheroids isolated from field-collected water sample.

[f] Sexual female spheroids with eggs isolated from field-collected water sample.

[g] Sexual female spheroids with matured zygotes isolated from field-collected water sample.

examined after mixing approximately 10 μL cultured material with 2–5 μL 0.002% (w/v in distilled water) methylene blue (1B-429 WALDECK GmbH & Co Division Chroma, Münster, Germany).

## Molecular experiments

To identify species and infer the phylogenetic position of the Japanese dioecious species, we used the internal transcribed spacer (ITS) regions of nuclear ribosomal DNA (rDNA; ITS-1, 5.8S rDNA, and ITS-2). The ITS rDNA sequences of cultured materials and field-collected sexual spheroids as well as two chloroplast genes (the large subunit of Rubisco [*rbcL*] and the photosystem II CP43 apoprotein [*psbC*] genes) of cultured materials were determined by direct sequencing of polymerase chain reaction (PCR) products as described previously [7] except for DNA template and enzyme reaction for PCR. For field-collected sexual spheroids, five morphologically identical male or female spheroids were washed with culture medium and pipetted into a tube into which 30 μL 2×PCR Buffer of KOD FX Neo (Toyobo, Osaka, Japan) and one ceramic bead (5 mm in diameter; YTZ ball, Nikkato Co., Osaka, Japan) were added. For clonal cultures, approximately 30 μL concentrated cultured material were mixed with 30 μL 2×PCR Buffer and a bead. Then the tubes were subjected to a Retsch Mixer Mill MM300 (F. Kurt Retsch GmbH & Co.KG, Haan, RP, Germany) with 30 Hz for 10 min to produce "disrupted cell solution" for template DNA for PCR. PCR reaction mixtures were prepared with the disrupted cell solution (10–15% v/v in the mixture) and KOD FX Neo (for ITS rDNA and *rbcL*) or KOD One PCR Master Mix (Toyobo) (for *psbC*), according to the manufacturer's protocol. The PCR schedule for *rbcL* and *psbC* was 2 min at 94˚C, followed by 45 cycles of 10 s at 98˚C, 30 s at 50˚C and 30 s at 68˚C. For PCR of ITS rDNA, the schedule was 2 min at 94˚C, followed by 40 cycles of 10 s at 98˚C, 30 s at 66˚C and 30 s at 68˚C.

For the phylogeny of ITS rDNA and *rbcL-psbC* [7], we analyzed the operational taxonomic units (OTUs) or species/samples/strains listed in Table 1. The sequences were aligned as described previously [3,4,7]. The alignments are available from TreeBASE (www.treebase.org/treebase-web/home.html; study ID: 24792). Designation of the root or outgroup was performed as in a previous study [7]. Maximum-likelihood (ML) analyses based on the ITS rDNA and *rbcL-psbC* alignments were performed using MEGA6.06 [12], with 1000 replicates of bootstrap analyses [13], In addition, Bayesian phylogenetic analyses for the respective alignments were carried out using MrBayes 3.2.6 [14], as described in a previous study [3]. The secondary structures of ITS-2 were predicted as described previously [2–4,7].

# Results

## Morphology of asexual spheroids of *Volvox rousseletii* from Japan

Mature asexual spheroids in culture were ovoid in shape with a broad posterior pole, measured 331–423 μm wide and 352–476 μm long, and contained 4700–11,800 cells embedded in individual sheaths at the periphery of the gelatinous matrix (Fig 1A and 1B). Somatic cells were connected to one another by cytoplasmic bridges thicker than flagella (Fig 1C), measuring up to 12 μm long. Somatic cells in the anterior region of the spheroid were pear-shaped to ovoid in side view, with the cell length longer than cell width (Fig 1D). Each somatic cell had two flagella, a single stigma, and a cup-shaped chloroplast with a single basal pyrenoid. Asexual spheroids typically had 4–8 gonidia. Gonidia were spherical in shape and distributed in the posterior two-thirds of the spheroid (Fig 1A). During daughter spheroid formation, gonidia of the next generation were evident in the embryo just after inversion (Fig 1E).

## Morphology of sexual spheroids of field-collected *Volvox rousseletii* from Japan

Mature male spheroids in field-collected samples were ovoid or ellipsoidal in shape, measured 304–468 μm wide and 337–534 μm long, and contained 9100–17,000 somatic cells and 35–92

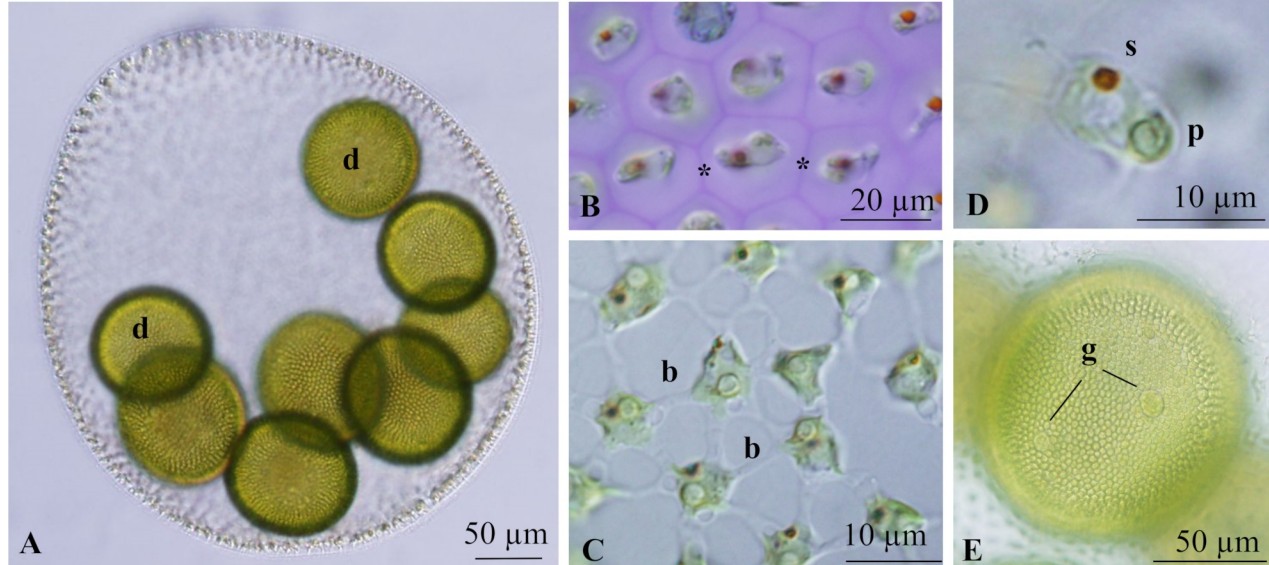

**Fig 1. Light microscopic features of asexual spheroids in culture of *Volvox rousseletii* strain v-sgm-17 from Lake Sagami, Japan (Table 1).** (A) Mature spheroid showing daughter spheroids (d). (B-D) Part of spheroids. (B) Top view of individual sheaths (asterisks) of somatic cells stained with methylene blue. (C) Top view of somatic cells with thick cytoplasmic bridges (b). (D) Side view of elongate-ellipsoidal, anterior somatic cell with stigma (s) and pyrenoid (p) in the chloroplast. (E) Developing embryo just after inversion, showing gonidia (g) of the next generation.

sperm packets (Fig 2A). Sperm packets were compressed globoids composed of biflagellate male gametes, 34–44 μm in diameter, and positioned in the posterior 2/3–3/4 of the colony (Fig 2B). The packets did not develop simultaneously; early stages of sperm packet development were present in old male spheroids (Fig 2A). Female spheroids with mature eggs or zygotes were also found (Fig 2C–2F). These were ovoid or ellipsoidal in shape, measured 363–550 μm wide and 380–657 μm long, and contained 8900–14,000 somatic cells and 52–97 eggs or zygotes distributed in the posterior 2/3 of the spheroids (Fig 2C and 2E). The eggs did not develop simultaneously, but fully matured female colonies contained only matured eggs or zygotes (Fig 2C and 2E). The mature zygote was spiny, and the spines were slightly curved or straight with acute apices (Fig 2F). Zygotes were 31–39 μm in diameter (excluding spines). Spines were 3.4–6.4 μm long.

## Induction of sexual spheroids of *Volvox rousseletii* from Japan in culture

Although the ITS sequences were identical between field-collected sexual spheroids and cultured materials, sexual spheroids induced in culture were not well developed, particularly in female strains; matured zygotes were not obtained even after mixing male and female spheroids induced in culture.

Induced male spheroids were essentially the same as those found in the water column except that they were smaller (Fig 3A and 3B). Fully matured sperm packets escaped from the parental male colony within the culture.

Female spheroids in culture were also smaller than those found in the lake. They contained only eggs (Fig 3C and 3D). The eggs in female strains sometimes had a smooth or spiny wall and became reddish brown in color to develop into parthenospores.

## Molecular identification and phylogeny

All specimens (excluding strains with nuclear rDNA ITS sequences corresponding to that of the monoecious species growing in the same lake, *Volvox* sp. Sagami [7]) exhibited identical

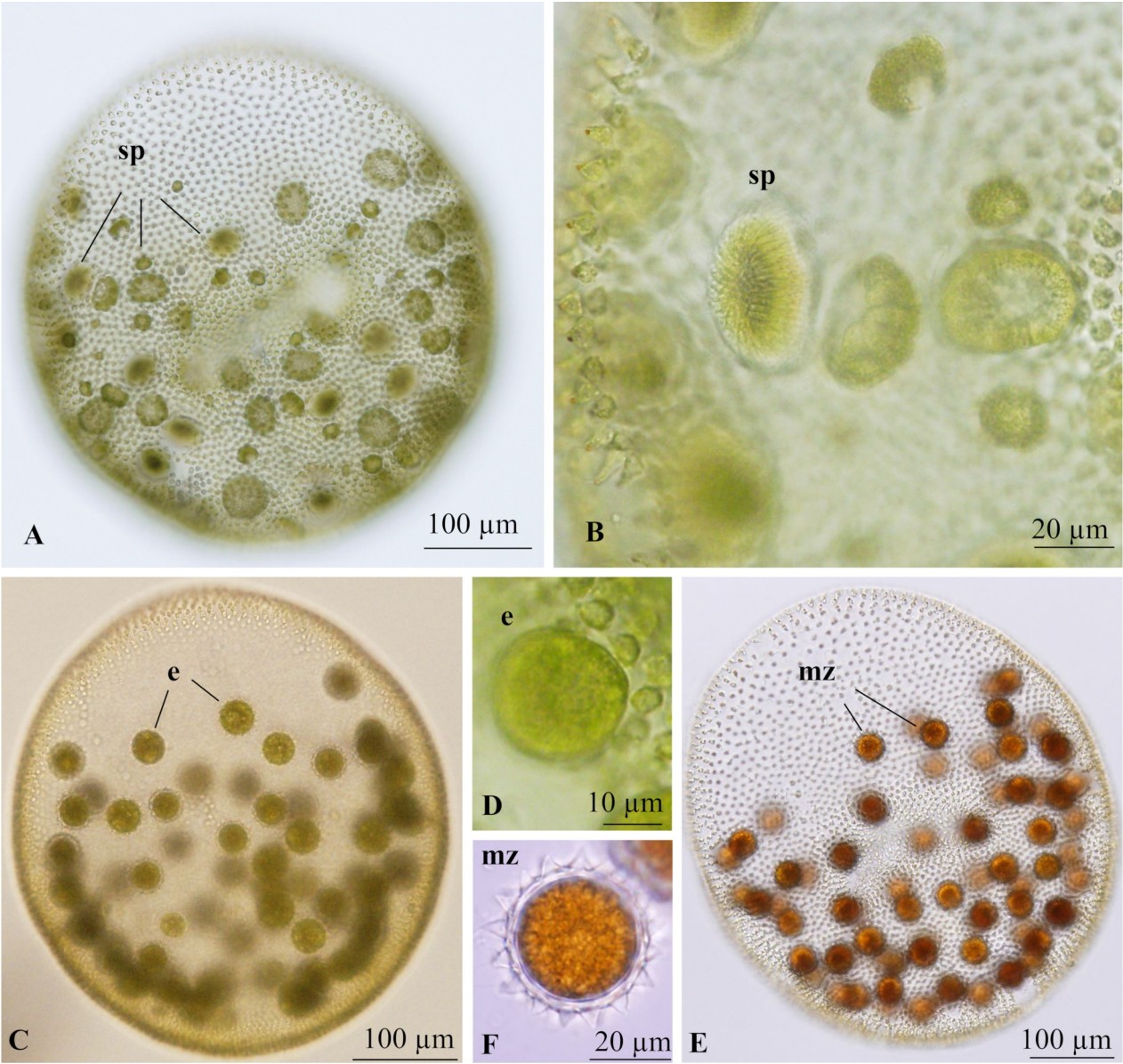

**Fig 2. Light microscopic features of sexual spheroids of *Volvox rousseletii* from field-collected samples in Lake Sagami, Japan (24 July 2018; Table 1).** (A) Male spheroid with sperm packets (sp) of various developing stages. (B) Sperm packets (sp) developing within male spheroid. (C) Female spheroid with eggs (e). (D) Side view of egg (e) in female spheroid. (E) Female spheroid with matured zygotes (mz). (F) Matured zygote (mz) with acute spines on zygote wall.

sequences in the nuclear rDNA ITS region. Figs 4 and 5 show the phylogenetic position of the dioecious species *Volvox rousseletii* from Japan, based on the ITS region and *rbcL-psbC* genes, respectively. The relationships were essentially the same as those of previous studies [5,7], except for the additional OTUs (*V. rousseletii* from Japan and *V. perglobator*). The Japanese dioecious species *V. rousseletii* was sister to *V. rousseletii* strain UTEX 1862 originating from South Africa [6] with 80–87% bootstrap values and 0.89–0.97 posterior probabilities. Based on comparisons of the secondary structure of ITS-2, no compensatory base changes were found between the Japanese and South African algae (S1 and S2 Figs).

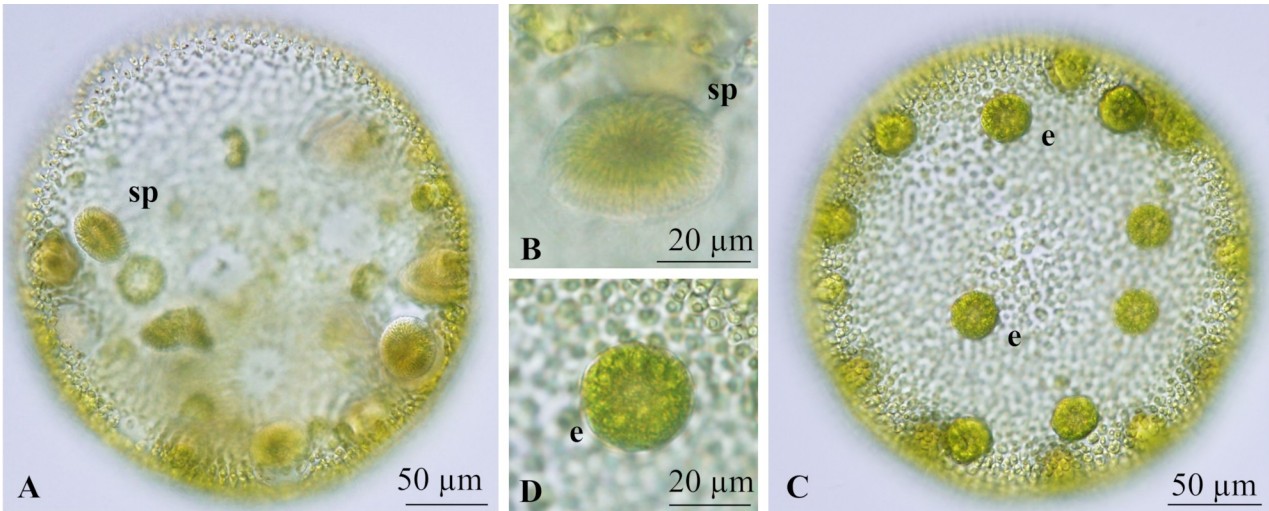

**Fig 3. Light microscopic features of sexual spheroids developing in culture of *Volvox rousseletii* v-sgm-24 (A, B) and v-sgm-23 (C, D) (Table 1).** (A) Male spheroid with sperm packets (sp). (B) Side view of sperm packet (sp) in male spheroid. (C) Female colony with eggs (e). (D) Egg (e) in female spheroid.

## Discussion

Three dioecious species of *Volvox* sect. *Volvox*, i.e., *V. rousseletii*, *V. perglobator*, and *V. prolificus*, can be clearly distinguished from one another based on differences in the morphology of sexual spheroids observed in field-collected materials [1,15,16]. However, the cultured materials in the present study did not produce adequately well-developed sexual spheroids for species identification (Fig 3). Thus, taxonomic data were obtained from the field-collected samples including male and female spheroids and mature zygotes (Fig 2); these samples were identified as the same species as the cultured materials using ITS sequences. Two species of *Volvox* sect. *Volvox* (monoecious *Volvox* sp. Sagami and dioecious *V. rousseletii*) generally grow simultaneously in Lake Sagami, and the asexual spheroids of the two species cannot be morphologically distinguished. Thus, morphological data for asexual spheroids of *V. rousseletii* described here were based on cultured materials (Fig 1) after molecular identification.

The present morphological data from the Japanese dioecious species of *Volvox rousseletii* are consistent with those from sexual spheroids and zygotes of *V. rousseletii* from Africa, in that they exhibit continuous development of sperm packets during the maturation of male spheroids, the nearly simultaneous maturation of eggs in female spheroids, and acute spines in the zygote wall (S1 Table). Although *V. rousseletii* var. *lucknowensis* was described based on Indian material [1, 16], spines of the zygote wall of this alga are broadly conical, representing a different species. In another dioecious species of this section, *V. perglobator* recorded only from the USA [1,5], sperm packet development in male spheroids is not continuous (old male spheroids contain only matured sperm packets), and spines of the zygotes are blunt (S1 Table). The remaining dioecious species of this section, *V. prolificus*, has only been recorded from India [1, 16]. In *V. prolificus*, eggs mature continuously from young to old stages of the parental female spheroid, and spines of the zygote walls are not acute [1, 16]. In addition, the present Japanese species was most closely related to *V. rousseletii* from South Africa in our phylogenetic analyses.

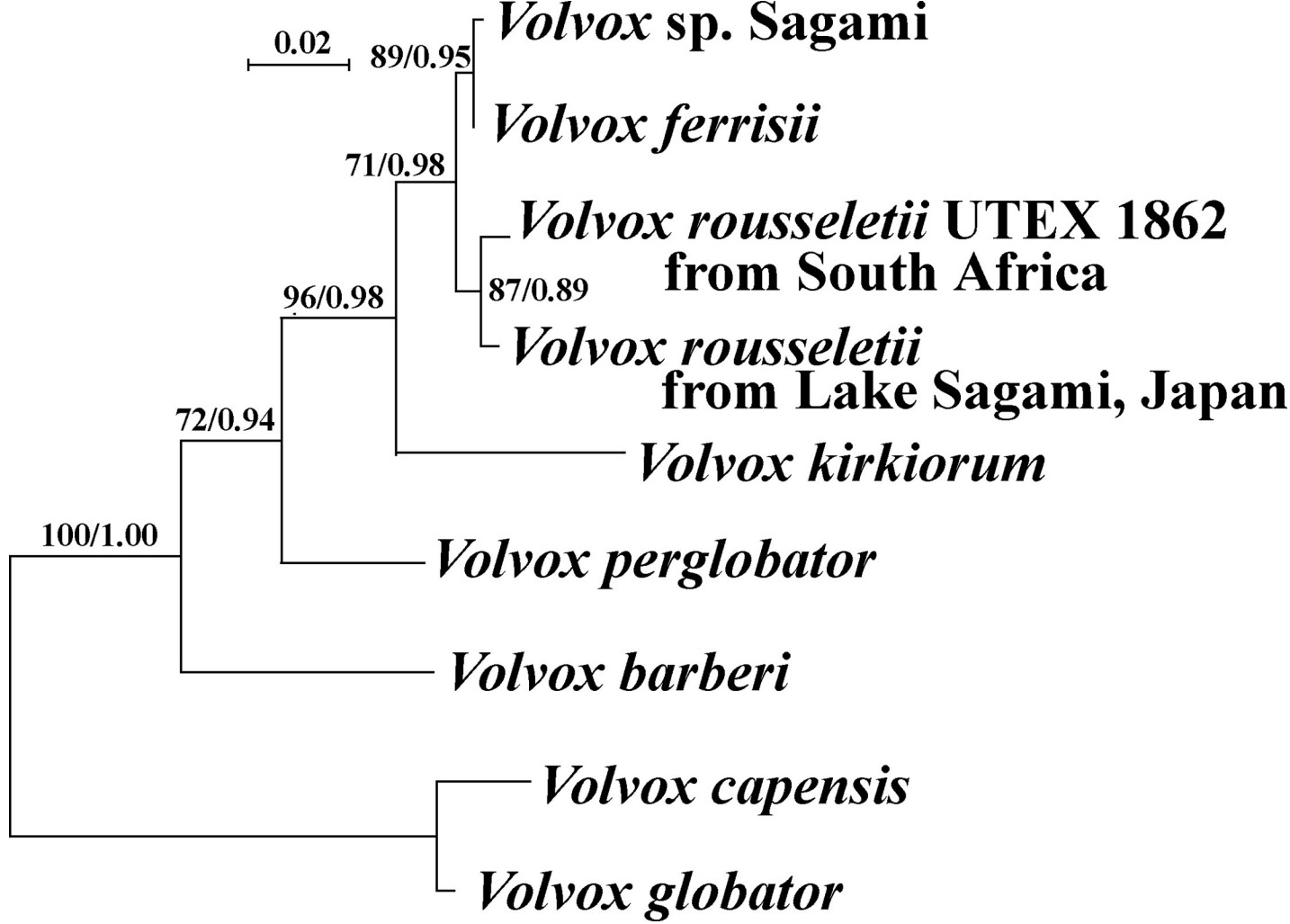

**Fig 4. Phylogenetic position of *Volvox rousseletii* originating from Japan (Table 1) within *Volvox* sect. *Volvox*, based on maximum likelihood (ML) analysis of the internal transcribed spacer (ITS) regions of nuclear ribosomal DNA (rDNA) (ITS-1, 5.8S rDNA, and ITS-2).** Branch lengths are proportional to the evolutionary distances that are indicated by the sale bar. Numbers in left and right sides at branches represent bootstrap values (50% or more) based on 1000 replications of ML and posterior probabilities (0.85 or more) by Bayesian inference, respectively.

## Conclusions

The present combined data set from field-collected and cultured materials of a *Volvox* species distributed in a Japanese lake demonstrated the occurrence of *V. rousseletii*, which has previously only been reported from Africa [1,6]. However, asexual and sexual spheroids of this Japanese alga are smaller than those of *V. rousseletii* reported from Africa (S1 Table). In addition, the Japanese alga is genetically different from the South African alga of the same species (Figs 4 and 5). Thus, the species range for *V. rousseletii* has been expanded based on the present study and more data are needed to resolve the biogeographical significance of the morphological and genetic variability within this species. Now, *V. rousseletii* is a unique dioecious species of *Volvox* sect. *Volvox* distributed in more than one continent, and fresh cultures of *V. tousseletii* originating from Japan and South Africa are available (Table 1). Additional dioecious species of *Volvox* sect. *Volvox* with intercontinental distribution and/or new species of this section may be revealed in further studies using both field-collected and cultured materials of this section growing in various freshwater habitats worldwide.

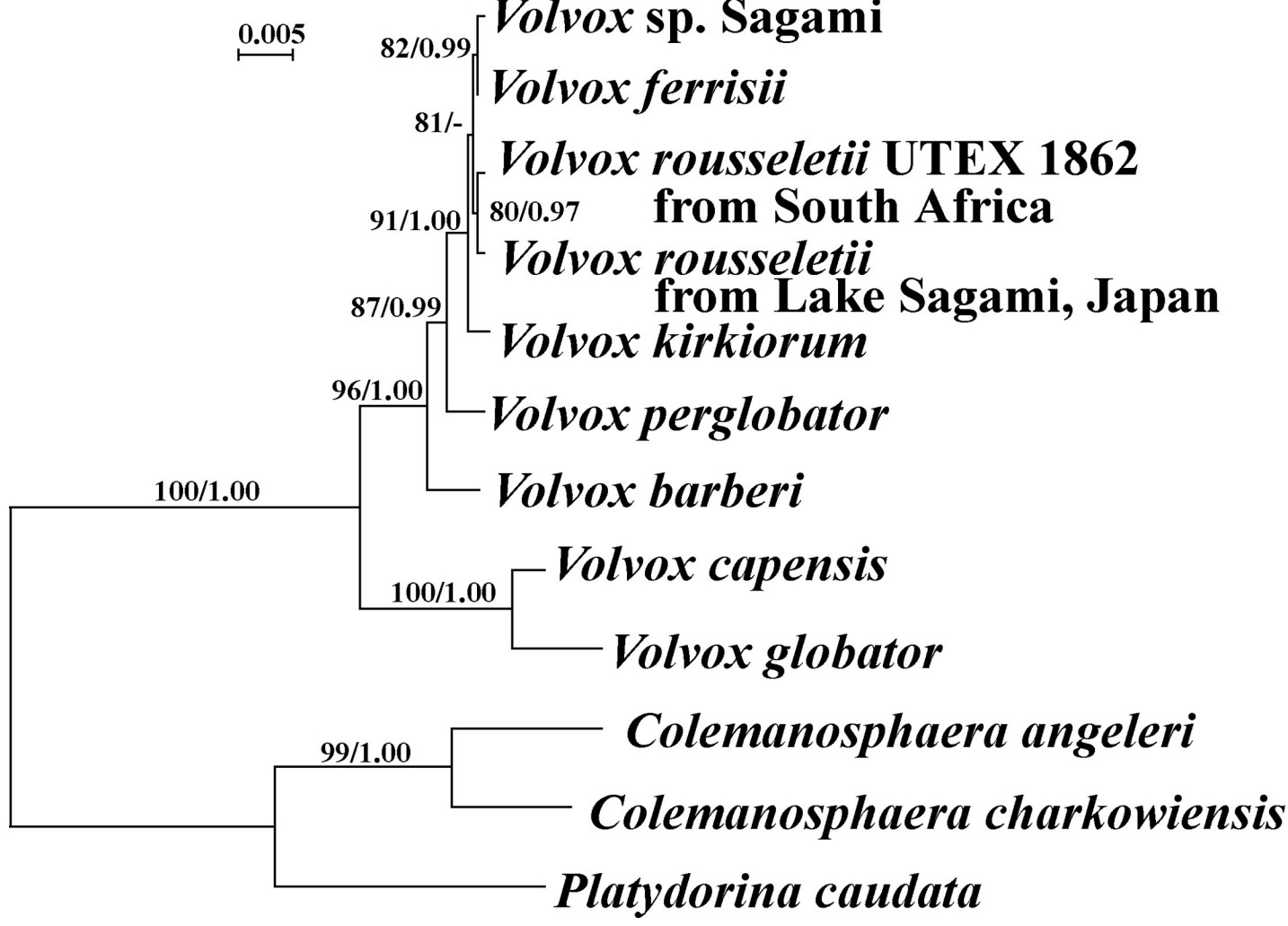

**Fig 5. Phylogenetic position of *Volvox rousseletii* originating from Japan (Table 1) within *Volvox* sect. *Volvox*, based on maximum likelihood (ML) analysis of combined data matrix of *rbcL* and *psbC* gene sequences. *Colemanosphaera* and *Platydorina* constitute outgroup.** Branch lengths are proportional to the evolutionary distances that are indicated by the sale bar. Numbers in left and right sides at branches represent bootstrap values (50% or more) based on 1000 replications of ML and posterior probabilities (0.85 or more) by Bayesian inference, respectively.

## Supporting information

**S1 Fig. The secondary structure of nuclear ribosomal DNA (rDNA) internal transcribed spacer 2 (ITS-2) transcript of *Volvox rousseletii* from Japan, including the 3' end of the 5.8S ribosomal RNA (rRNA) and the 5' end of the large subunit of rRNA (LSU rRNA).** (DOCX)

**S2 Fig. Comparison of helices of the secondary structure of nuclear ribosomal DNA internal transcribed spacer 2 transcripts between *Volvox rousseletii* from Japan and its related strain/species (Figs 4 and 5).** (DOCX)

**S1 Table. Comparison of dioecious species originating from Japan and previously described dioecious species of *Volvox* sect. *Volvox*.** (DOCX)

## Acknowledgments

We are grateful to Ms. Shizue Arii and Ms. Izumi Tateno (Kanagawa Prefecture, Japan), who helped us in the field collections. Copyright. 2019 Kimbara et al. This is an open access article distributed under the terms of the Creative Commons Attribution License, which permits unrestricted use, distribution, and reproduction in any medium, provided the original author and source are credited.

## Author Contributions

**Conceptualization:** Nanako Isaka.

**Data curation:** Ryosuke Kimbara, Ryo Matsuzaki, Hisayoshi Nozaki.

**Formal analysis:** Ryosuke Kimbara, Ryo Matsuzaki.

**Funding acquisition:** Hisayoshi Nozaki.

**Investigation:** Ryosuke Kimbara, Nanako Isaka, Ryo Matsuzaki, Hiroko Kawai-Toyooka, Hisayoshi Nozaki.

**Methodology:** Ryosuke Kimbara, Nanako Isaka, Ryo Matsuzaki, Hiroko Kawai-Toyooka, Hisayoshi Nozaki.

**Project administration:** Hisayoshi Nozaki.

**Resources:** Masanobu Kawachi.

**Supervision:** Masanobu Kawachi, Hisayoshi Nozaki.

**Writing – original draft:** Ryosuke Kimbara, Ryo Matsuzaki, Hisayoshi Nozaki.

**Writing – review & editing:** Ryosuke Kimbara, Nanako Isaka, Ryo Matsuzaki, Hiroko Kawai-Toyooka, Masanobu Kawachi, Hisayoshi Nozaki.

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
