## [Decision Letter · Decision Letter 0]

16 Jul 2019

PONE-D-19-17716

Morphological and Molecular Identification of the Dioecious “African Species” Volvox rousseletii (Chlorophyceae) in the Water Column of a Japanese Lake Based on Field-collected and Cultured Materials

PLOS ONE

Dear Dr. Nozaki,

Thank you for submitting your manuscript to PLOS ONE. After careful consideration, we feel that it has merit but does not fully meet PLOS ONE’s publication criteria as it currently stands. Therefore, we invite you to submit a revised version of the manuscript that addresses the points raised during the review process.

I agree with Reviewer 1 about the high quality and overall suitability of this manuscript for publication.  In your revision and rebuttal please respond to the critiques of Reviewer 1 with whom I also agree regarding the last section where the discussion of Spirogyra and Chloromonas seemed out of place, and the need for details of the KOD polymerase method to be elaborated in the Methods section.

One more minor point. In Fig. 1B there are asterisk symbols which are not mentioned in the legend.

We would appreciate receiving your revised manuscript by Aug 30 2019 11:59PM. To enhance the reproducibility of your results, we recommend that if applicable you deposit your laboratory protocols in protocols.io, where a protocol can be assigned its own identifier (DOI) such that it can be cited independently in the future. For instructions see: http://journals.plos.org/plosone/s/submission-guidelines#loc-laboratory-protocols

We look forward to receiving your revised manuscript.

Kind regards,

James G. Umen, Ph. D.

Academic Editor

PLOS ONE

3. Thank you for stating the following in the Acknowledgments Section of your manuscript: "Funding: HN was supported by a Grant-in-Aid for Scientific Research (grant number 16H02518) from the Ministry of Education, Culture, Sports, Science and Technology (MEXT)/Japan Society for the Promotion of Science (JSPS) KAKENHI (https://www.jsps.go.jp/english/e-grants/). The funder had no role in study design, data collection and analysis, decision to publish, or preparation of the manuscript."

Please remove any funding-related text from the manuscript and let us know how you would like to update your Funding Statement. Currently, your Funding Statement reads as follows:  "The funders had no role in study design, data collection and analysis, decision to publish, or preparation of the manuscript."

Additional Editor Comments (if provided):

Reviewers' comments:

Reviewer's Responses to Questions

**Comments to the Author**

1. Is the manuscript technically sound, and do the data support the conclusions?

Reviewer #1: Yes

2. Has the statistical analysis been performed appropriately and rigorously? 

Reviewer #1: Yes

3. Have the authors made all data underlying the findings in their manuscript fully available?

Reviewer #1: Yes

4. Is the manuscript presented in an intelligible fashion and written in standard English?

Reviewer #1: Yes

5. Review Comments to the Author

Reviewer #1: This manuscript details the occurrence of Volvox rousseletii in Japan, the first time this species has been recorded outside of Africa. The manuscript is well written and is both concise and clear. The data presented is micrographs of laboratory cultivated asexual colonies, field collected sexual colonies, and phylogenetic analyses deriving from both asexual and sexual colonies (confirming that the sexual colonies do indeed correspond to the associated asexual strains). Overall the analyses are appropriate, the data recording is meticulous, the micrographs are excellent, and the paper should be published.

I have one major comment which should be addressed before publication. While the manuscript title, Abstract, and Introduction emphasizes the characterization of a species thought to be restricted to Africa, the Conclusions section instead emphasizes the challenge of characterizing sexual isolates in culture in Spirogyra and Chloromonas, other algae not previously mentioned. The Conclusions should be revised to address the novelty of this paper, the isolation of V. rousseletii in Japan as emphasized by the title, Abstract, and Introduction. The revised species range for V. rousseletii and availability of fresh cultures is worthwhile and should be emphasized. Similarly, the usage of KOD polymerase, an important methodological detail, should be moved to the Methods.

A few minor comments:

1. Hanschen et al 2018 Evol Eco Res published the psbC and rbcL sequences of Volvox perglobator (accession numbers KY489659 and KY489662), these should be included in Figure 5.

2. Page 5, last paragraph “Volvox” should be italicized when introducing VTAC media

3. The implementation of maximum parsimony for phylogenetic analyses is outdated and should be replaced by Bayesian analyses. This analysis can be implemented in MrBayes.

4. Page 9, please do not abbreviate CBCs.

5. Figure 1, the (*) and (b) are not specified in the legend for panels B and C.

6. PLOS authors have the option to publish the peer review history of their article (what does this mean?). If published, this will include your full peer review and any attached files.

Reviewer #1: No

---

## [Author Response · Author response to Decision Letter 0]

31 Jul 2019

PONE-D-19-17716

Morphological and Molecular Identification of the Dioecious “African Species” Volvox rousseletii (Chlorophyceae) in the Water Column of a Japanese Lake Based on Field-collected and Cultured Materials

PLOS ONE

Dear Dr. Nozaki,

Thank you for submitting your manuscript to PLOS ONE. After careful consideration, we feel that it has merit but does not fully meet PLOS ONE’s publication criteria as it currently stands. Therefore, we invite you to submit a revised version of the manuscript that addresses the points raised during the review process.

I agree with Reviewer 1 about the high quality and overall suitability of this manuscript for publication. In your revision and rebuttal please respond to the critiques of Reviewer 1 with whom I also agree regarding the last section where the discussion of Spirogyra and Chloromonas seemed out of place, and the need for details of the KOD polymerase method to be elaborated in the Methods section.

Responses: Based on the comments, the conclusion section has been drastically revised by deleting the discussion of Spirogyra and Chloromonas. Detailed methods of PCR with KOD polymerase have been described in Materials and Methods section of the revised manuscript. 

One more minor point. In Fig. 1B there are asterisk symbols which are not mentioned in the legend.

Response: The asterisk symbols have been specified in the legend for Fig. 1B in the revised manuscript.

We would appreciate receiving your revised manuscript by Aug 30 2019 11:59PM. To enhance the reproducibility of your results, we recommend that if applicable you deposit your laboratory protocols in protocols.io, where a protocol can be assigned its own identifier (DOI) such that it can be cited independently in the future. For instructions see: http://journals.plos.org/plosone/s/submission-guidelines#loc-laboratory-protocols

• A rebuttal letter that responds to each point raised by the academic editor and reviewer(s). This letter should be uploaded as separate file and labeled 'Response to Reviewers'.

• A marked-up copy of your manuscript that highlights changes made to the original version. This file should be uploaded as separate file and labeled 'Revised Manuscript with Track Changes'.

• An unmarked version of your revised paper without tracked changes. This file should be uploaded as separate file and labeled 'Manuscript'.

We look forward to receiving your revised manuscript.

Kind regards,

James G. Umen, Ph. D.

Academic Editor

PLOS ONE

Response: We have ensured that our revised manuscript meets PLOS ONE's style requirements

3. Thank you for stating the following in the Acknowledgments Section of your manuscript: "Funding: HN was supported by a Grant-in-Aid for Scientific Research (grant number 16H02518) from the Ministry of Education, Culture, Sports, Science and Technology (MEXT)/Japan Society for the Promotion of Science (JSPS) KAKENHI (https://www.jsps.go.jp/english/e-grants/). The funder had no role in study design, data collection and analysis, decision to publish, or preparation of the manuscript."

Please remove any funding-related text from the manuscript and let us know how you would like to update your Funding Statement. Currently, your Funding Statement reads as follows: "The funders had no role in study design, data collection and analysis, decision to publish, or preparation of the manuscript."

Responses: I have removed any funding-related text from the revised manuscript. We would not like to update our Funding Statement "The funders had no role in study design, data collection and analysis, decision to publish, or preparation of the manuscript."

Responses: I have included captions for our Supporting Information files at the end of our revised manuscript and updated any in-text citations to match accordingly. I have seen your Supporting Information guidelines

Additional Editor Comments (if provided):

Reviewers' comments:

Reviewer's Responses to Questions

Comments to the Author

1. Is the manuscript technically sound, and do the data support the conclusions?

Reviewer #1: Yes

2. Has the statistical analysis been performed appropriately and rigorously? 

Reviewer #1: Yes

3. Have the authors made all data underlying the findings in their manuscript fully available?

Reviewer #1: Yes

4. Is the manuscript presented in an intelligible fashion and written in standard English?

Reviewer #1: Yes

5. Review Comments to the Author

Reviewer #1: This manuscript details the occurrence of Volvox rousseletii in Japan, the first time this species has been recorded outside of Africa. The manuscript is well written and is both concise and clear. The data presented is micrographs of laboratory cultivated asexual colonies, field collected sexual colonies, and phylogenetic analyses deriving from both asexual and sexual colonies (confirming that the sexual colonies do indeed correspond to the associated asexual strains). Overall the analyses are appropriate, the data recording is meticulous, the micrographs are excellent, and the paper should be published.

I have one major comment which should be addressed before publication. While the manuscript title, Abstract, and Introduction emphasizes the characterization of a species thought to be restricted to Africa, the Conclusions section instead emphasizes the challenge of characterizing sexual isolates in culture in Spirogyra and Chloromonas, other algae not previously mentioned. The Conclusions should be revised to address the novelty of this paper, the isolation of V. rousseletii in Japan as emphasized by the title, Abstract, and Introduction. The revised species range for V. rousseletii and availability of fresh cultures is worthwhile and should be emphasized. Similarly, the usage of KOD polymerase, an important methodological detail, should be moved to the Methods.

Responses: Based on the comments, the conclusion section has been drastically revised by deleting the discussion of the challenge of characterizing sexual isolates in culture in Spirogyra and Chloromonas. Revised species range for V. rousseletii and availability of fresh cultures from Japan have been discussed in the conclusion section of the revised manuscript. Detailed methods of PCR with KOD polymerase have been described in Materials and Methods section of the revised manuscript. 

A few minor comments:

1. Hanschen et al 2018 Evol Eco Res published the psbC and rbcL sequences of Volvox perglobator (accession numbers KY489659 and KY489662), these should be included in Figure 5.

Responses: Based on the comment, the psbC and rbcL sequences Volvox perglobator (accession numbers KY489659 and KY489662) have been included in Figure 5 of the revised manuscript.

2. Page 5, last paragraph “Volvox” should be italicized when introducing VTAC media

Response: Done as suggested.

3. The implementation of maximum parsimony for phylogenetic analyses is outdated and should be replaced by Bayesian analyses. This analysis can be implemented in MrBayes.

Response: The maximum parsimony has been replaced by Bayesian analyses using MrBayes in Figures 4 and 5 of the revised manuscript.

4. Page 9, please do not abbreviate CBCs.

Response: Done as suggested.

5. Figure 1, the (*) and (b) are not specified in the legend for panels B and C.

Response: The (*) and (b) have been specified in the legend for panels B and C in the revised manuscript.

---

## [Editor Report · Decision Letter 1]

13 Aug 2019

Morphological and Molecular Identification of the Dioecious “African Species” Volvox rousseletii (Chlorophyceae) in the Water Column of a Japanese Lake Based on Field-collected and Cultured Materials

PONE-D-19-17716R1

Dear Dr. Nozaki,

We are pleased to inform you that your manuscript has been judged scientifically suitable for publication and will be formally accepted for publication once it complies with all outstanding technical requirements.  Congratulations on adding an interesting new set of isolates to the Volvox family.

Note that the revised manuscript file set did not include Table S1 which I assume did not change from the first submission. Please remember to send this file to the PLoS ONE editorial staff.

With kind regards,

James G. Umen, Ph. D.

Academic Editor

PLOS ONE
---

## [Editor Report · Acceptance letter]

22 Aug 2019

PONE-D-19-17716R1 

Morphological and Molecular Identification of the Dioecious “African Species” *Volvox rousseletii* (Chlorophyceae) in the Water Column of a Japanese Lake Based on Field-collected and Cultured Materials 

Dear Dr. Nozaki:

I am pleased to inform you that your manuscript has been deemed suitable for publication in PLOS ONE. Congratulations! Your manuscript is now with our production department. 

With kind regards,

on behalf of

Dr. James G. Umen 

Academic Editor

PLOS ONE